# *GLUL* Ablation Can Confer Drug Resistance to Cancer Cells via a Malate-Aspartate Shuttle-Mediated Mechanism

**DOI:** 10.3390/cancers11121945

**Published:** 2019-12-05

**Authors:** Magesh Muthu, Ranjeet Kumar, Azharuddin Sajid Syed Khaja, Jonathan D. Gilthorpe, Jenny L. Persson, Anders Nordström

**Affiliations:** 1Department of Molecular Biology, Umeå University, 90187 Umeå, Sweden; magesh.muthu@umu.se (M.M.); ranjeetmku09@gmail.com (R.K.); sk.azharuddin@umu.se (A.S.S.K.); jenny.persson@umu.se (J.L.P.); 2Department of Pharmacology and Clinical Neuroscience, Umeå University, 90187 Umeå, Sweden; jonathan.gilthorpe@umu.se

**Keywords:** *GLUL*, NSCLC, drug resistance, metabolomics, glutamine, glycolysis, metabolism, targeted metabolomics, LC-MS

## Abstract

Glutamate-ammonia ligase (GLUL) is important for acid-base homeostasis, ammonia detoxification, cell signaling, and proliferation. Here, we reported that *GLUL* ablation conferred resistance to several anticancer drugs in specific cancer cell lines while leaving other cell lines non-resistant to the same drugs. To understand the biochemical mechanics supporting this drug resistance, we compared drug-resistant *GLUL* knockout (KO) A549 non-small-cell lung carcinoma (NSCLC) cells with non-resistant *GLUL* KO H1299 NSCLC cells and found that the resistant A549 cells, to a larger extent, depended on exogenous glucose for proliferation. As GLUL activity is linked to the tricarboxylic acid (TCA) cycle via reversed glutaminolysis, we probed carbon flux through both glycolysis and TCA pathways by means of ^13^C5 glutamine, ^13^C5 glutamate, and ^13^C6 glucose tracing. We observed increased labeling of malate and aspartate in A549 GLUL KO cells, whereas the non-resistant GLUL KO H1299 cells displayed decreased ^13^C-labeling. The malate and aspartate shuttle supported cellular NADH production and was associated with cellular metabolic fitness. Inhibition of the malate-aspartate shuttle with aminooxyacetic acid significantly impacted upon cell viability with an IC_50_ of 11.5 μM in resistant GLUL KO A549 cells compared to 28 μM in control A549 cells, linking resistance to the malate-aspartate shuttle. Additionally, rescuing GLUL expression in A549 KO cells increased drug sensitivity. We proposed a novel metabolic mechanism in cancer drug resistance where the increased capacity of the malate-aspartate shuttle increased metabolic fitness, thereby facilitating cancer cells to escape drug pressure.

## 1. Introduction

Cancer cells undergo a process of continuous clonal evolution, which increases the chances of selecting for drug resistance during therapy [1,2,3]. Although recent therapeutic advances have improved the specificity of cancer treatment, drug resistance is an ongoing and serious problem for patient outcome [4]. Resistance mechanisms can manifest as alterations in the drug target, activation of pro-survival pathways, or ineffective induction of cell death. However, there is no drug available to target resistance mechanisms, per se. Overexpression of the multidrug resistance gene (*ABCB1*), encoding the permeability glycoprotein P-gp has been attributed to ineffective treatment [5,6,7] and directly implicated in treatment failure [8,9,10]. Targeting P-gp with a number of novel inhibitors has not yet translated into clinical utility due to toxicity and pharmacokinetic complications [11]. Identification of novel mechanisms that support cancer cell survival under therapeutic pressure is important in order to find ways of overcoming drug resistance, which ultimately results in lethality.

While deregulated cellular metabolism appears to be an emerging hallmark of cancer drug resistance [12,13,14,15], deregulated energetics involving alterations in cellular metabolism to support neoplastic proliferation is a fundamental hallmark of cancer [16]. Central to this metabolic deregulation is Warburg’s discovery of aerobic glycolysis, where glucose is converted to lactate even when oxygen supply is sufficient to support oxidative phosphorylation. However, several studies have shown that aerobic glycolysis alone is insufficient to support the core metabolism required for cancer cell proliferation [17,18]. The tricarboxylic acid (TCA) cycle is a metabolic intersection that connects cellular energy metabolism with more distant metabolic pathways through key nodes for anaplerosis/cataplerosis. One such node is α-ketoglutaric acid, which can be supplied through a glutaminolysis mechanism, whereby glutamine is first converted to glutamate by the action of glutaminase (GLS). Glutamate conversion, for example, by glutamate dehydrogenase (GLUD) generates α-ketoglutaric acid. It is well known that many cancer cell types are dependent on increased flux from glutamine to glutamate and α-ketoglutaric acid [19]. Hence, upregulated glutamine metabolism is an attractive target for cancer therapy, as evidenced by an ongoing trial with the glutaminolysis inhibitor CB-839 (ClinicalTrials.gov Identifier: NCT03428217 and NCT02771626).

Glutamate-ammonia ligase (GLUL) catalyzes the ATP-dependent synthesis of glutamine from glutamate and ammonia, which is the reverse of the initial glutaminolysis step. GLUL plays an important role in glutamate and ammonia detoxification and acid-base homeostasis. GLUL was shown to be involved in pathological cell migration during angiogenesis [20]. Thus, GLUL could be important beyond its main catalytic activity during tumor growth [20]. Increased GLUL expression has been reported as an early marker of hepatocellular carcinoma [21], promoting breast cancer cell proliferation [22], and also found to be an unfavorable prognostic marker in patients with glioblastoma multiforme (GBM) and ovarian cancer [23,24]. Our group has previously identified reduced *GLUL* transcription to be associated with resistance to the chemotherapeutic agent daunorubicin in clones of acute lymphoblastic leukemia (ALL) [14]. This finding prompted us to examine if a targeted reduction of GLUL expression could induce drug resistance. We investigated the effect of reduced GLUL expression using siRNA or lentiviral CRISPR-Cas9 mediated knockout (KO), as well as doxycycline-inducible shRNA-mediated knockdown (KD) in different cancer cell lines. Interestingly, *GLUL* KO/KD resulted in a gain of function phenotype with induced drug resistance in specific cancer cell types, including the non-small cell lung cancer (NSCLC) cell line A549. Metabolic profiling and stable isotope-labeled tracer experiments showed that resistance was supported through increased glucose dependence coupled with increased activity in the malate-aspartate shuttle, which is a mechanism for transporting electrons into mitochondria and thus fueling regeneration of NADH from NAD+. The activity of the malate-aspartate shuttle has been associated with longevity in yeast [25] and supports up to 20% of the respiration rate in various tumor types [26]. Here, we demonstrated that pharmacological inhibition of the malate-aspartate shuttle reduced viability in resistant *GLUL* KO A549 cells compared to control cells, thus connecting malate-aspartate metabolism with drug tolerance in cancer cells. Furthermore, re-expression of *GLUL* in KO cells restored the sensitivity of cells to drug treatment, suggesting that the expression level of *GLUL* might influence drug sensitivity in specific cancer cell types. Since the genetic loss of function of catalytic enzymes rarely results in a gain of function phenotype, our data suggested that the level of *GLUL* expression could fine-tune metabolic fitness, which in turn might offer therapeutic opportunities for combination therapies targeting metabolic fitness during induction treatment in order to suppress selection of resistant clones.

## 2. Results

### 2.1. Transient GLUL Knockdown Induces Drug Resistance

We previously observed that drug-resistant ALL cells lacked *GLUL* transcription [14]. In the present study, we explored whether or not reduced GLUL expression resulted in drug resistance in solid tumor-derived cell lines. We examined GLUL protein levels by western blotting in a panel of cancer cell lines, including A549, H1299, H460 (NSCLC), HeLa (cervical cancer), HCC1954 (breast ductal carcinoma), MDA-MB-231 (triple-negative breast cancer—TNBC). A relatively high level of GLUL expression was found in HeLa cells compared to the other lines (Figure 1A). To test whether *GLUL* KD could induce drug resistance, we first evaluated the effectiveness of siRNA-mediated *GLUL* KD by western blot analysis. After 72 h of siRNA transfection, there was a profound decrease in GLUL protein expression in all of the cell lines tested (Figure 1B). Cells were then treated with the chemotherapeutic agent docetaxel (20 or 30 nM for 72 h), and the cell viability was assessed by MTS assay. Interestingly, knocking down *GLUL* promoted drug resistance in two of the cell lines (A549 and HCC1954; Figure 1C). As *GLUL* KD induced the highest level of drug resistance in A549 cells but had no apparent effect in the NSCLC H1299 cells, we chose to compare these two cell lines further to identify potential resistance mechanisms.

### 2.2. GLUL Knockout Cells Display Drug Resistance

We next used a lentiviral CRISPR/Cas9 system to generate complete and stable *GLUL* KOs in NSCLC A549 and H1299 cells. These cell lines constitute a good experiment/control pair as they are both of NSCLC origin while demonstrating resistance/non-resistance phenotype upon *GLUL* KO. The approaches yielded clones that were devoid of GLUL protein expression (Appendix A). Two weeks after transduction, independent clones were selected and propagated. For A549, we utilized clones 1 and 4 for further investigation (Figure 2A, Appendix A). *GLUL* KO A549 and control lines were exposed to various chemotherapeutic agents, including dasatinib, pazopanib, imatinib, docetaxel, and cisplatin. Cell viability was then determined by means of MTS assay. Exposure to each of the compounds in a dose-dependent manner resulted in a significant loss of viability in control A549 cells, and IC_50_ values more than doubled in A549 *GLUL* KO cells (Figure 2B, Appendix A). To characterize the broad spectrum of drug resistance in *GLUL* KO A549 cells further, we performed clonogenic assays. Cells were grown in the presence of dasatinib (5 µM), pazopanib (10 µM), imatinib (10 µM), cisplatin (5 µM), and docetaxel (3 nM) for 12 days. Control cells displayed a reduced ability for colony formation when compared to the *GLUL* KOs (Figure 2C, Appendix A), further demonstrating enhanced drug resistance in the *GLUL* KO cells.

To confirm that *GLUL* loss of function potentiates drug resistance in A549 cells, we established stable cell lines expressing doxycycline-inducible shRNAs (shRNA 1, 2, 3) that target *GLUL* (Appendix A). *GLUL* shRNA3 was significant in its ability to ablate GLUL protein expression upon doxycycline induction, and individual clones capable of expressing shRNA 3 were selected and expanded (Appendix A). We first examined the effect of various chemotherapeutic drugs, including dasatinib, pazopanib, and imatinib, on A549 cells expressing non-target or negative control shRNAs in comparison to A549 *GLUL* shRNA3 clone3 (cl3) (*GLUL* KD) cells through MTS assay. After 72 h of treatment, dose-response curves showed that 10–20 μM of each drug induced a significant loss of cell viability in A549 non-target or control cells, whereas in A549 *GLUL* KD cells, twice the dose of each of these drugs was required to achieve 50% of cell growth inhibition (Appendix A).

In order to differentiate the downstream effects of *GLUL* KO with respect to resistance, we also generated stable *GLUL* KO in non-resistant H1299 cells using the CRISPR-Cas9 method. Loss of GLUL protein expression was significant in H1299 cells, and lines expanded from individual clones were analyzed (Figure 2A). Additionally, we examined the cell viability in the presence of various pharmacological drugs in both H1299 control and H1299 *GLUL* KO Clone 2 (cl2) cells and observed no significant difference with their growth inhibition after 72 h (Figure 2B, Appendix A). Also, we performed clonogenic assays. Cells were grown in the presence of pazopanib (10 µM), docetaxel (5 nM), and dasatinib (3 µM) for 12 days. There was no significant difference in colony formation between control and *GLUL* KO cells (Figure 2C, Appendix A). This suggested that the ability of *GLUL* KO to confer drug resistance was confined to specific cancer cell types.

### 2.3. GLUL KO-Associated Drug Resistance Correlates with Apoptosis

We next investigated the cellular pathway of chemotherapeutic drug resistance associated with *GLUL* KO in A549 cells in relation to apoptosis. Cleavage of poly(ADP-ribose) polymerase 1 (PARP-1) is a prominent marker of early apoptosis [27,28]. Western blot analysis showed that 10 and 20 µM of pazopanib and imatinib; 20 and 30 nM of docetaxel, as well as 5 and 10 µM of dasatinib, stimulated PARP cleavage in A549 control cells. In contrast, *GLUL* KO cells displayed no, or reduced, PARP cleavage after 12 h treatment (Figure 3). Consistently, we observed the same effect following *GLUL* KD in A549 shRNA3 cl3 cells (Appendix A). Our results showed that *GLUL* loss of function mediated a poor response to chemotherapeutic drugs by downregulating the apoptotic pathway in A549 cells.

### 2.4. GLUL Knockout Cells Display Glucose Dependence

Previously, we have shown that drug-resistant ALL cells with reduced *GLUL* transcription were characterized by reduced glutamine and increased glucose dependency [14]. Therefore, we explored the effects of reducing the concentrations of glutamine and glucose in the media on cell proliferation using MTS assay. No relative difference was observed between *GLUL* KO A549 and H1299 cells in proliferation under reduced media glutamine levels (Appendix A). However, reducing the glucose concentration impacted significantly the proliferation of drug-resistant *GLUL* KO A549 cells compared to drug-sensitive A549 control cells (Figure 4). To understand this phenomenon, we treated both A549 and H1299 (control/ko) cells with 5 mM of 2-deoxy-D-glucose (2-DG), a glycolysis inhibitor [29] under various concentrations of glucose, and the cell viability was assessed through MTS assay. Indeed, A549 KO cells displayed higher sensitivity to 2-DG treatment with respect to their controls. This effect was not observed for H1299 cells (Figure 4). Thus, the drug resistance phenotype appeared to be connected with an increased requirement for glucose or increased dependence on glycolytic function.

### 2.5. A Metabolic Phenotype Involving the Malate-Aspartate Shuttle Supports Drug Resistance Associated with GLUL KO

To separate the resistance phenotype from the effects of *GLUL* KO, we compared two *GLUL* KO lines A549 (resistant) with H1299 (non-resistant). We performed targeted relative-quantitative profiling of 80 different metabolites in both A549 and H1299 cells. Several metabolites were uniquely associated with significant differences in only one of the cell line pairs when comparing KO to the negative control (Appendix A). Interestingly, glutamine accumulated when *GLUL* was knocked down, regardless of the drug resistance phenotype (Appendix A). It would be expected that a reduced capacity for de novo glutamine synthesis in the *GLUL* KO cell lines would affect glutamine uptake and/or the rate of glutaminolysis (conversion of glutamine to glutamate and α-ketoglutarate). Indeed, the glutamine transporter SLC1A5 (Solute Carrier Family 1 Member 5) and the glutamate synthesizing enzyme, GLS, both showed increased expression in *GLUL* KD conditions (Appendix A).

This prompted us to probe glutamine uptake and glutaminolysis flux by feeding cells with ^13^C_5_ labeled glutamine. Quantifying the %-fractional labeling after 24 h revealed how much of the intracellular pool of a metabolite was labeled with a stable isotope through uptake or biosynthesis (Appendix A). A small quantitative difference in glutamine levels between KO and control cells was observed (Appendix A). For glutamate, no difference between the cell lines was found, indicating that glutamine uptake was slightly increased when GLUL was reduced. However, the conversion of glutamine to glutamate was not altered following *GLUL* KO. The reduced %-fractional labeling of α-ketoglutarate was observed in both KO cell lines, suggesting that a reduced rate of glutaminolysis was mediated through reduced GLUL activity. As this was observed in both resistant and non-resistant cells, it could be concluded that altered glutaminolysis flux does not support resistance. When cells were exposed to either ^13^C_5_ glutamate or ^13^C_5_ glucose, a similar reduction in the labeling of α-ketoglutarate was observed (Appendix A). Interestingly, de novo glutamine synthesis was barely detectable in any of the cell lines as the fraction of labeled glutamine in the presence of ^13^C_5_ glutamate only represented about 1%. This low level was reduced still further in KO cells (Appendix A), suggesting that the direct catalytic activity of GLUL was not essential for sustaining intracellular glutamine levels in these cells.

To investigate the fate of apparently increased glucose utilization in resistant *GLUL* KO A549 cells (Figure 4), we followed the %-fractional labeling of glycolytic and TCA intermediates using ^13^C_6_-labeled glucose and LC-MS. We were specifically interested in finding labeling patterns that differed between the resistant A549 *GLUL* KO cells and the non-resistant H1299 *GLUL* KO cells. Labeling of glycolytic intermediates showed similarly trending patterns for both cell lines as well as the TCA intermediates α-ketoglutarate and citrate (Figure 5). However, malate and aspartate displayed a different pattern, with increased fractional labeling in A549 *GLUL* KO cells and reduced labeling in H1299 *GLUL* KO cells (Figure 5). An altered labeling pattern in malate and aspartate could imply that the malate-aspartate shuttle was linked to the drug-resistant phenotype. Increased flux through the malate aspartate shuttle could work together with increased glucose utilization towards increasing metabolic fitness through increased capacity for NADH production.

### 2.6. Inhibition of the Malate-Aspartate Shuttle Reveals Metabolic Vulnerability Associated with Resistance

The malate-aspartate shuttle is associated with increased metabolic fitness in a number of cancer cell types [26] and has been implicated in cancer cell proliferation [30,31]. Therefore, we hypothesized that *GLUL* ablation could mediate an increased capacity of the malate-aspartate shuttle. Thus, even in the absence of chemotherapeutic drugs, *GLUL* KO cancer cells should be more dependent on the support of this shuttle for mitochondrial NADH production and more sensitive to its inhibition. We tested this hypothesis by application of aminooxyacetic acid to inhibit the aspartate aminotransferase (AAT) (Figure 5), an enzyme that supports the malate-aspartate shuttle [32]. Aminooxyacetic acid had an IC_50_ of 11.5 μM in resistant *GLUL* KO A549 cells (Figure 6) and an IC_50_ of 28 μM in non-resistant control A549 cells. No such effects were observed in non-resistant H1299 *GLUL* KO cells (Appendix A). These results provided evidence for a mechanism of drug resistance in *GLUL* ablated cells that were mechanistically supported through increased dependence on glycolysis linked with higher dependence on the malate-aspartate shuttle.

### 2.7. Rescuing GLUL Expression Sensitizes A549 KO Cells to Drugs 

To verify that *GLUL* KO is responsible for the drug-resistant phenotype, we verified that *GLUL* sensitization to drug treatment could be rescued by GLUL re-expression in *GLUL* KO A549 cells. A Flag-tagged GLUL expression cassette was introduced in KO cells, and GLUL expression levels were analyzed by western blotting. Subsequently, A549 KO expressing Flag-*GLUL* and A549 KO cells were treated with docetaxel (20 nM) for 12 h and were analyzed for apoptosis by annexin V staining. Flag-GLUL re-sensitized A549-KO cells to docetaxel (Figure 6), confirming that GLUL expression influenced drug responsiveness in a specific cancer cell type.

## 3. Discussion

With the advancement in our understanding of cancer metabolism, new opportunities for more efficient and less toxic cancer therapies will develop by exploiting the metabolic vulnerabilities of cancer cells that are not present in normal proliferating cells [33]. Targeting the mechanisms of drug resistance is a focal point in this endeavor, as these mechanisms will enable certain cancer cells to survive initial therapy and ultimately cause disease relapse. Recent findings from our lab have suggested that drug-resistant ALL cells lack *GLUL* transcription [14]. Importantly, we showed here that *GLUL* ablation conferred drug resistance in A549 NSCLC cells, and that this resistance was supported mechanistically through modulation of glycolytic flux linked with an upregulation of malate-aspartate transport in mitochondria. This shuttle is associated with increased metabolic fitness by enhancing the capacity of the cellular respiration [25,26]. Finally, targeting AAT as an essential component of the malate-aspartate shuttle [32,34] with aminooxyacetic acid had a larger impact on the viability of drug-resistant A549 *GLUL* KO cells and their drug-sensitive control counterparts. These findings supported a mechanistic model where *GLUL* ablation increased both glycolytic flux and the malate-aspartate shuttle concurrently. This increased the metabolic fitness of cells, aiding cell survival under drug selective pressure.

Recent studies in vivo have shown that the role of glutamine metabolism in cancer may be more complicated than previously understood [35]. The type of oncogene associated with a tumor, as well as the localization of the tumor tissue, appears to dictate the level of glutamine anaplerosis/cataplerosis in relation to the TCA cycle [36]. Furthermore, a mouse NSCLC model did not catabolize glutamine through glutaminolysis at all but depended solely on glucose-derived carbon flux [37]. Several studies have revealed a prognostic value of *GLUL* expression in different cancer types [21,22,23,24]. Further, the ablation of *GLUL* in SK-BR-3 cells reduced their proliferation [22]. Here, we showed that knockdown of *GLUL* induced drug resistance in A549 (NSCLC) and HCC1954 (breast cancer) cells (Figure 1A–C). Consistently, *GLUL* KD and/ or KO in A549 cells displayed drug resistance in a cell viability assay. Additionally, both *GLUL* KD and KO also displayed reduced or no PARP cleavage in response to a range of anticancer drugs (Figure 3, Appendix A), supporting the conclusion that *GLUL* ablation suppressed apoptosis under selective pressure. Thus, mutations leading to reduced *GLUL* transcription should provide a survival advantage for cancer cells. To support this prediction, we queried the public database resource Kaplan–Meier plotter [38], which combines mRNA expression with patient survival data. We found strong significance for a correlation between reduced survival and lower *GLUL* mRNA expression in both breast and lung cancer patient cohorts (Appendix A).

The multidrug-resistant protein (MDR1, ABCB1 or P-glycoprotein) has been associated with drug failure in patients [8,9]. The expression of this protein was not regulated by *GLUL* ablation in our study (Appendix A). Nor could we correlate the rate of cell growth to drug resistance, consistent with our western blot analysis of proliferating cell nuclear antigen (PCNA), a well-known cellular marker for proliferation [39] (Appendix A). These findings suggest that the resistance is not driven by an increased expression of drug efflux pump or by reduced proliferation.

Previously, we observed altered dependence on exogenous glucose and glutamine for proliferation in resistant ALL cells lacking *GLUL* [14]. In the present study, we observed an increased dependence on glucose for proliferation in drug-resistant A549 *GLUL* KO (Figure 4), suggesting that cellular energetics were involved in driving the resistance phenotype. Interestingly *GLUL* KO cells did not show glutamine dependence that could be associated with a resistance phenotype (Appendix A). To pinpoint changes in metabolism that could underlie the observed drug resistance phenotype, we used targeted metabolic profiling screening in *GLUL* KO/control pairs of A549 and H1299 cells (Appendix A). Metabolic profiling data revealed that both A549 and H1299 *GLUL* KO cells accumulated glutamine. This observation was supported mechanistically by increased SLC1A5 expression upon doxycycline-induced *GLUL* KD (Appendix A). However, stable isotope labeling did not result in measurable increased ^13^C_5_-glutamine uptake over 24 h (Appendix A). Additionally, flux from ^13^C_5_-glutamine to ^13^C_5_-glutamate (glutaminolysis) was analyzed by calculation of the %-fractional labeling of ^13^C in glutamate (Appendix A). There was no increase in labeled glutamate in A549 or H1299 *GLUL* KO cells, suggesting that altered glutaminolysis did not directly drive the observed drug resistance mechanism. This was further supported by measuring a proportionally similar decrease in %-fractional labeling in α-ketoglutarate in both resistant and sensitive A549 and H1299 *GLUL* KO cells (Appendix A).

With the apparently increased requirement of glucose for proliferation (Figure 4) in A549 *GLUL* KO cells, we probed labeling through glycolysis and the TCA cycle using ^13^C_6_-glucose to gauge if there was a discrepancy in the %-fractional labeling intermediate metabolites between A549 and H1299 *GLUL* KO cells. Distinctively, the TCA intermediate malate displayed increased labeling in A549 *GLUL* KO cells and reduced labeling in H1299 *GLUL* KO cells compared to their respective controls. The same labeling patterns were also observed for aspartate (Figure 5). Malate and aspartate are coupled in a shuttle-facilitating transport of electrons into the mitochondrial matrix from the intermembrane space/cytosol (i.e., the malate-aspartate shuttle), enhancing cellular respiration [40,41]. This shuttle provides increased metabolic fitness in various cells for proliferation [26,30,31]. In addition, the cataplerotic function of TCA cycle intermediates for a range of other metabolite classes, including those involved in lipid and nucleotide biosynthesis [12,42], suggesting a wider potential role for increased flux in malate and aspartate that may represent a critical point in metabolic rewiring, which mediates drug resistance in A549 *GLUL* KO cells.

Previous studies have indicated that pharmacological targeting of the malate-aspartate shuttle can inhibit cancer cell proliferation [30,31]. Administration of aminooxyacetic acid to target aspartate aminotransferase [34], which plays an integral part in the malate-aspartate shuttle (Figure 5), revealed that resistant A549 *GLUL* KO cells were significantly more sensitive to drug treatment compared to A549 control cells; illustrated with both viability and a colony-forming assay (Figure 6). Furthermore, a recent study reported that *GLUL* overexpression sensitized NSCLCs to anti-cancer drug gefitinib treatment [43]. Consistent with that finding, rescuing expression of GLUL in A549 KO cells re-sensitized cells to docetaxel treatment with respect to their controls (Figure 6), suggesting that *GLUL* might play a vital role in determining drug sensitivity in targeted therapies.

## 4. Material and Methods

### 4.1. Cell Culture and Reagents

NSCLCs A549 (ATCC^®^ CCL-185™, Manassas, VA, USA), H1299 (ATCC^®^ CRL-5803™), H460 (ATCC^®^ HTB-117™), the cervical cancer cell line HeLa (ATCC^®^ CCL-2™), and breast cancer cell lines HCC1954 (ATCC^®^ CRL-2338™) and MDA-MB-231 (ATCC^®^ HTB-26™) were acquired from the American Type Culture Collection and maintained according to the ATCC’s recommendations. DMEM (41965039), MEM (10370047), RPMI-1640 (72400021), and FBS (16000044) were obtained from Thermofisher Scientific (Thermofisher Scientific, Waltham, MA, USA). Cell culture media were supplemented with 10% (*v/v*) fetal bovine serum (FBS), 100 units/mL of penicillin, and 100 μg/mL of streptomycin. Cells were cultivated in a humidified atmosphere at 37 °C and 5% (*v/v*) CO_2_.

FDA approved chemotherapeutic drugs: dasatinib (approved for chronic myeloid leukemia and ALL), pazopanib (multi-targeted tyrosine kinase inhibitor, approved for renal cell carcinoma (RCC) [44] and soft tissue sarcoma [45]), docetaxel (inhibitor of microtubule formation [46], approved for breast cancer, head and neck cancer, stomach cancer, prostate cancer, and NSCLC), and cisplatin (induces apoptosis via DNA damage [47], approved for testicular cancer, ovarian cancer, cervical cancer, breast cancer, bladder cancer, head and neck cancer, esophageal cancer, lung cancer, mesothelioma, brain tumors, and neuroblastoma) were purchased from Selleckchem (Houston, TX, USA).

Aminooxyacetic acid (C13408) was obtained from Sigma Aldrich (D2650, Sigma Aldrich, St. Louis, MO, USA), dissolved in dimethyl sulfoxide (DMSO) at a stock concentration of 50 mM, and stored at −80 °C. Anti-GLUL (ab64613), anti-GLS (ab93434), anti-PCNA (ab29), and anti-α-tubulin (ab4074) antibodies were purchased from Abcam (Cambridge, UK). Anti-GLUL (LS-C114787) was purchased from LifeSpan Biosciences Inc. (Seattle, WA, USA). Anti-cleaved PARP (5625), anti-MDR1 (12683), anti-AKT (4685), and anti-pAKT (4060) were from Cell Signaling Technology (Beverly, MA, USA). Secondary antibodies: HRP-conjugated anti-mouse (1721011), anti-rabbit (1721019) were from Bio-Rad (Hercules, CA, USA). Amersham ECL western blotting detection reagent (RPN2106) was purchased from GE Healthcare Lifesciences (Pittsburgh, PA, USA).

### 4.2. GLUL siRNA Transfection

Cells were seeded in 6-well plates (2 × 10^5^ per well) for 24 h before transfection. After 12–18 h, cells were transfected with either scrambled RNA (20 nM)—(ON-TARGETplus Control pool–Non-Targeting pool–D-001810-10-05—Dharmacon, Lafayette, CO, USA) or *GLUL* siRNA (20 nM)—(ON-TARGET plus SMARTpool–Human *GLUL*–L-008228-01-0010—Dharmacon) for 72 h before treatment.

### 4.3. Establishment of Inducible GLUL KD Cell Lines by shRNA Expression

To knockdown *GLUL* expression, three individual SMARTvector Inducible Human *GLUL* shRNAs were designed as follows; shRNA1–V3sH7669-225458080 (GTCCGGATAGCTACGCCTA), shRNA2–V3SH7669-226004989 (GTTACAATCGGGACAACTG) targeting the 3ʹUTR, shRNA3–V3SH7669-228686437 (TAGACGTCGGGCATTGTCC) targeting the ORF, and a Non-Target shRNA (VSC6584) as a negative control was purchased from Dharmacon. For lentiviral *GLUL*-shRNA transduction, the concentration of multiplicity of infection (MOI) 2.0 was optimized for each *GLUL*-shRNA expressing GFP, and the complete medium was added to HEK293T cells for 72 h; then, the supernatant medium was collected, sterile filtered, and applied to A549 cells with 8 µg/mL polybrene and incubated for 72 h. Subsequently, fresh media containing 10 µg/mL puromycin (ant-pr-1; Invivogen, San Diego, CA, USA) were replaced after 48 h. From this pool of transduced cells, single cells expressing GFP were isolated for clonal expansion by FACS analysis.

To induce *GLUL* knockdown, 2.0 × 10^5^ cells passaged in media containing tetracycline-free FBS were seeded into each well of a 6-well plate, and 5 µg/mL doxycycline was added after 12 h. Cells were analyzed after 48–96 h.

### 4.4. Establishment of Stable GLUL KO Cell Lines by CRISPR/Cas9

*GLUL* KO cells were generated using CRISPR/Cas9-mediated deletion. A transfer plasmid lentiCRISPRv2 with guide RNA1 (TGTTTCGGGACCCCTTCCGT)—Transcript: *GLUL*-201 ENST00000311223.9—falls in exon 4 (coding sequence/translation start site is in exon 3), and guide RNA2 (TATTACTGTGGTGTGGGAGC)—Transcript: *GLUL*-201 ENST00000311223.9—falls in exon 6 (coding sequence) were co-transfected into HEK293T cells with packaging plasmids pVSVg (8454; Addgene, Watertown, MA, USA) and psPAX2 (12260; Addgene) [48,49]. As a negative control, LentiCRISPR V2 negative control/human non-targeting guide RNA-(Forward primer: 5ʹ-ACGGAGGCTAAGCGTCGCAA-3ʹ, Reverse primer: 5ʹ-TTGCGACGCTTAGCCTCCGT-3ʹ; from GeCKO v2 libraries) was co-transfected into HEK293T cells with packaging plasmids pVSVg and psPAX2 using Lipofectamine LTX reagent (Thermo Fisher Scientific, Waltham, MA, USA). After 48 h, virus-containing supernatant was collected and filtered (0.45 µM). This was added to cells together with 8 µg/mL polybrene and complete medium. After incubating for 48 h, transduced cells were selected with 10 µg/mL puromycin. After 1 week of antibiotic selection, single cells were sorted by FACS analysis for clonal expansion.

### 4.5. GLUL Re-Expression in A549 GLUL KO Cells

*GLUL* KO cells were seeded a day before transfection in 6-well plates (3 × 10^5^ per well). After 12 h, cells were transfected with either *GLUL*-ORF cDNA clone-NM_001033044.3 (constructed with pcDNA3.1+/C-(K)-DYK vector) or empty vector (pcDNA3.1+/C-(K)-DYK), purchased from Genscript (Piscataway, NJ, USA). For transfection, 5 µg empty vector or *GLUL*-ORF cDNA clone, as well as Lipofectamine 3000, were diluted with Opti-MEM. After 5 min of incubation, diluted DNA and Lipofectamine 3000 were combined and incubated for 15–20 min to allow DNA-Lipofectamine 3000 complexes to form. The complexes were added with complete medium to cells and incubated. After 72 h of transfection, cells were subjected to apoptosis assay and western blot analysis.

### 4.6. MTS Assay and Western Blot Analysis

Cell growth inhibition was analyzed by MTS Colorimetric Assay Kit (ab197010; Abcam). Cells were seeded in 96-well plate (5 × 10^3^ per well) in six replicates and then treated with chemotherapeutic drugs, including dasatinib, imatinib, pazopanib, docetaxel (1 nM to 50 nM), cisplatin, 2-DG (50 mM), and aminooxyacetic acid at various concentrations (500 nM to 300 µM) for 48–96 h. Negative controls were treated with 0.1% (*v/v*) DMSO in serum-free medium. After drug treatment, cells were treated with 20 µL MTS reagent per well for 1–3 h, and then the absorbance was measured at 490 nm (Tecan Infinite 200 PRO, Männedorf, Switzerland). The background signal was normalized with the signal from adjacent wells containing medium and MTS reagent.

Western blot analyses were performed according to standard procedures. For protein extraction, cells were lysed with RIPA buffer (R0278; Sigma Aldrich) containing Protease Inhibitor Cocktail (P8340; Sigma Aldrich) and Phosphatase Inhibitor Cocktail (P2850; Sigma Aldrich). Extracted proteins were separated by SDS-PAGE using 10% polyacrylamide gels and then transferred to a PVDF membrane (Millipore, Darmstadt, Germany). After blocking in 5% milk or bovine serum albumin in Tris-buffer saline-Tween 20 (0.1%) (TBS-T), blots were probed with primary antibodies and then secondary antibodies. Antibody binding was visualized using ECL reagent according to the manufacturer’s instructions (GE Healthcare Life Sciences, Pittsburgh, PA, USA) following exposure to X-ray film (Agfa Healthcare NV, Mortsel, Belgium). The blots were re-probed with an antibody against α-tubulin as a loading control.

### 4.7. Clonogenic Assay

Cells were seeded into each well of a 6-well plate (2 × 10^3^ per well), and after 12 h, treated with drugs, including dasatinib (5 or 3 µM), pazopanib (10 µM), imatinib (10 µM), cisplatin (5 µM), and docetaxel (3 nM) in complete medium. Cells were grown for 12 days; then, cells were fixed with methanol/acetic acid (5:1) for 5 min and stained with crystal violet 0.5% solution (S25275B; Fisher Scientific, Hampton, NH, USA) for 30 min. The total number of colonies per well were counted and compared to negative controls. 

### 4.8. Flow Cytometry (FACS)-Based Apoptosis Analysis

Flow cytometry (FACS)-based assays were performed on A549 GLUL KO cells treated with docetaxel to analyze apoptosis. For apoptosis assay, PE-conjugated annexin V and 7-AAD staining was performed according to the manufacturer’s protocol (BD Biosciences, San Jose, CA, USA). Data were acquired using BD LSR II flow cytometer, and the analysis was performed with FCS Express 6Plus software (De Novo Software, Pasadena, CA, USA).

### 4.9. Metabolic Profiling and Stable Isotope Labeling and Tracing

For targeted metabolomics, cells were seeded into a 6-well plate (2 × 10^5^ cells/well) and grown for 12 h. Cells were then incubated with stable isotopes for 24 h and washed at least once with PBS, scraped, and transferred to a 1.5 mL Eppendorf tube. The cells were lysed, and metabolites were extracted using 200 μL 90:10 methanol:H_2_O containing 0.36 μM ^13^C_9_-phenylalanine as an internal standard, to which acid-washed glass-beads 425–600 μM from Sigma Aldrich (St. Louis, MO, USA) were added to constitute ~50% *v/v* of the 90:10 MeOH:H_2_O cell suspension and the cells were disrupted by shaking at 30 Hz for 2 min (Mixer Mill MM400, Retsch, Germany) using pre-chilled holding blocks (4 °C). Samples were centrifuged at 4 °C/14,000 rpm for 15 s, after which 100 µL of water was added, and samples were shaken again for 1 min for 15 s before the centrifugation step was repeated. Subsequently, 150 µL of the supernatant was transferred to LC-MS glass vials, from which 2–10 μL was injected into an Agilent 1290 LC-system connected to either a 6550 or 6560 Agilent Q-TOF mass spectrometer (Agilent, Santa Clara, CA, USA). An electrospray ionization (ESI) source was used in all of the LC-MS measurements. Data were collected between m/z 70 to 1700 in positive/negative ion mode. The following ESI settings were used (Agilent Jetstream): gas temperature 300 °C, gas flow 8 l/min, nebulizer pressure 40 psi, sheet gas temperature 350°C, sheet gas flow 11, Vcap 4000, fragmentor 100, Skimmer1 45, and OctapoleRFPeak 750. Metabolites were separated using reversed-phase chromatography (HSS T3 C18, 50 × 2.1 mm, 1.8 µM 100 Å (Waters, MA, USA)) or HILIC (iHILIC-Fusion (+), 100 × 2.1 mm, 3.5 µM, 100 Å, Hilicon AB (Umeå, Sweden)). For reversed-phase chromatography elution, solvents were (A) H_2_O, 0.1% formic acid and (B) 75:25 acetonitrile:isopropanol, 0.1% formic acid. For separation, the following linear gradient was used (flow rate 0.5 mL/min), min 0: 0.1% B, min 2: 10% B, min 7: 99% B, min 9: 99% B, min 9.3: 0.1% B, min 10.8: 0.1% B. Column re-equilibration occurred from 9.8 to 10.7 min with an increase in flow rate to 0.8 mL/min. HILIC elution solvents were (A) H_2_O, 50 mM ammonium formate (B) 90:10 (acetonitrile):(H_2_O 50 mM ammonium formate), total concentration ammonium formate in B 5 mM. The chromatographic separation was achieved using the following linear gradient (flow rate 0.4 mL/min), min 0: 90% B; min 4: 85% B, min 5: 70% B, min 7: 55% B, min 10: 20% B, min 10.01: 90% B, min 15: 90% B. Raw data were processed using Masshunter Profinder and the function batch target feature extraction (Agilent). Target feature libraries were obtained using synthetic standards in an in-house generated PCDL library (Agilent) through recording LC retention times and accurate masses. The internal standard ^13^C_9_-phenylalanine was used for analytical quality monitoring and detected in all four separation/ionization modes. Ratios for KO/control were calculated for all metabolite responses that were different (*p* < 0.05) between KO and control (ctrl) cells.

For stable isotope labeling, cells were seeded in a 6-well plate (2.0 × 10^5^ cells/well) and grown overnight. After 12–18 h, complete DMEM containing 10% FBS and stable isotopes, including ^13^C_5_-Glutamic acid (604860; Sigma Aldrich, St. Loius, MO, USA)–500 µM, ^13^C_5_-Glutamine (605166; Sigma Aldrich)–500 µM, and ^13^C_6_-D-Glucose (CLM-1396-1; PR-22544; Lardon Fine Chemicals AB, Sweden)–12.5 mM, were added to cells with adjustment of the concentration of unlabeled glutamine or glucose in the culture medium to maintain 4 mM and 25 mM, respectively. After 24 h of stable isotope incubation, cells were washed with PBS and subjected to metabolite extraction and LC-MS analysis, as described above. The %-fractional labeling was calculated through data processing using the Batch Isotopologue Extraction function in Agilent Masshunter Profinder (B 08.00, Agilent).

### 4.10. Statistical Analysis

Statistical analysis was performed using GraphPad Prism software 7 (San Diego, CA, USA). All the mean values are the average value of all samples. The standard deviation (SD) is an indication of the variability of all samples. The sample mean is indicated by the standard error. The confidence level is expressed using a 95% confidence interval (CI). All of the statistical tests were two-sided, and *p* < 0.05 was considered to be statistically significant. For all in vitro experiments, at least three independent experiments were conducted. All quantification data presented in the study was an average of at least three independent experiments. All western blots represented are at least out of two independent experiments. All IC_50_ curves are nonlinear regression fit plotted against log (inhibitor) vs. response (three parameters) using Graphpad Prism software (San Diego).

## 5. Conclusions

Here, we showed that *GLUL* ablation induced resistance to several different cancer drugs in specific cell lines. This gain of function phenotype provided an advantage to cancer cells under therapeutic pressure. Drug-resistant cells depended more on exogenous glucose for proliferation. Stable isotope-labeled glucose provided evidence for increased flux through malate and aspartate in resistant cells. We interpreted this as an increased reliance on the malate-aspartate shuttle in resistant cells, and that the enhanced metabolic fitness that follows mechanistically supports the resistance phenotype and aids cells in their escape from therapeutic pressure.

## Figures and Tables

**Figure 1 cancers-11-01945-f001:**
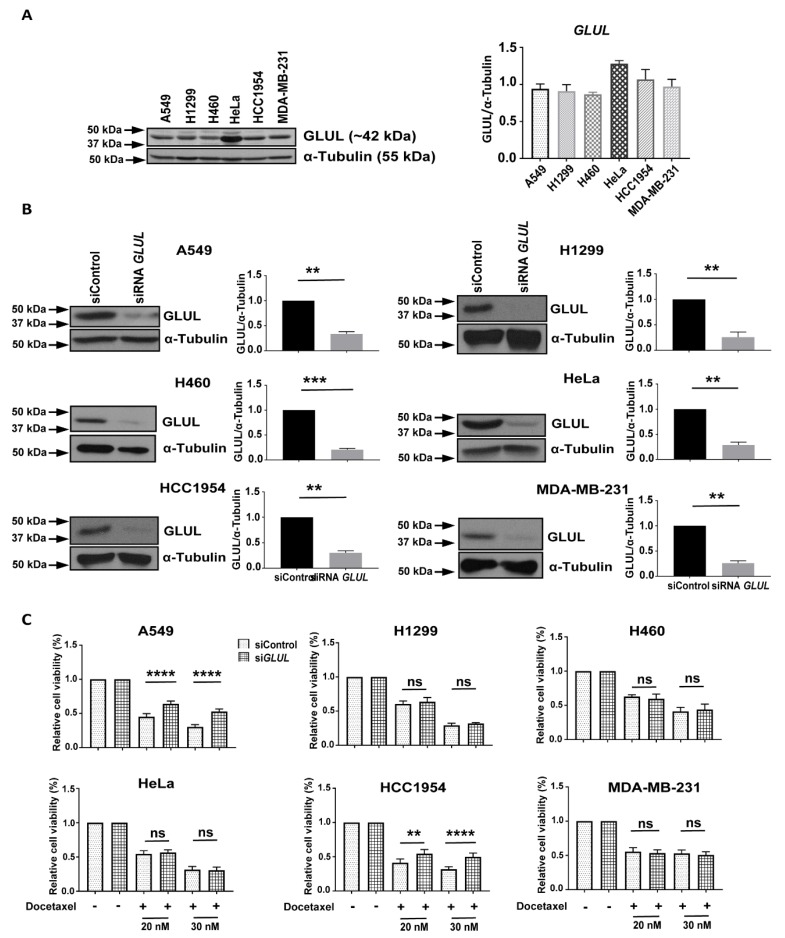
Reduced *GLUL* expression induced drug resistance. (**A**) GLUL (glutamate-ammonia ligase) protein expression was analyzed in different cancer cell lines. (**B**) Cell lines were either transfected with scrambled (siControl) or with siRNA *GLUL* as noted, and levels of GLUL protein expression were analyzed by western blotting. The western blot membranes were subsequently probed with an anti-tubulin antibody to assess equal loading. The presence of GLUL and tubulin protein is indicated on the right side of each blot. The approximate location of various molecular weight markers is indicated on the left side of each blot. kDa, kilo Dalton. (**C**) *GLUL* knockdown cell lines were treated with 20 and 30 nM of docetaxel for 72 h, and the cell viability was quantified by MTS assay. The standard error (SE) bars in cell viability graphs represent means of three independent experiments. The data are shown as mean ± SEM; *p*-values were determined using a two-tailed unpaired t-test; **** *p* ≤ 0.0001; ** *p* ≤ 0.002; ns—not significant.

**Figure 2 cancers-11-01945-f002:**
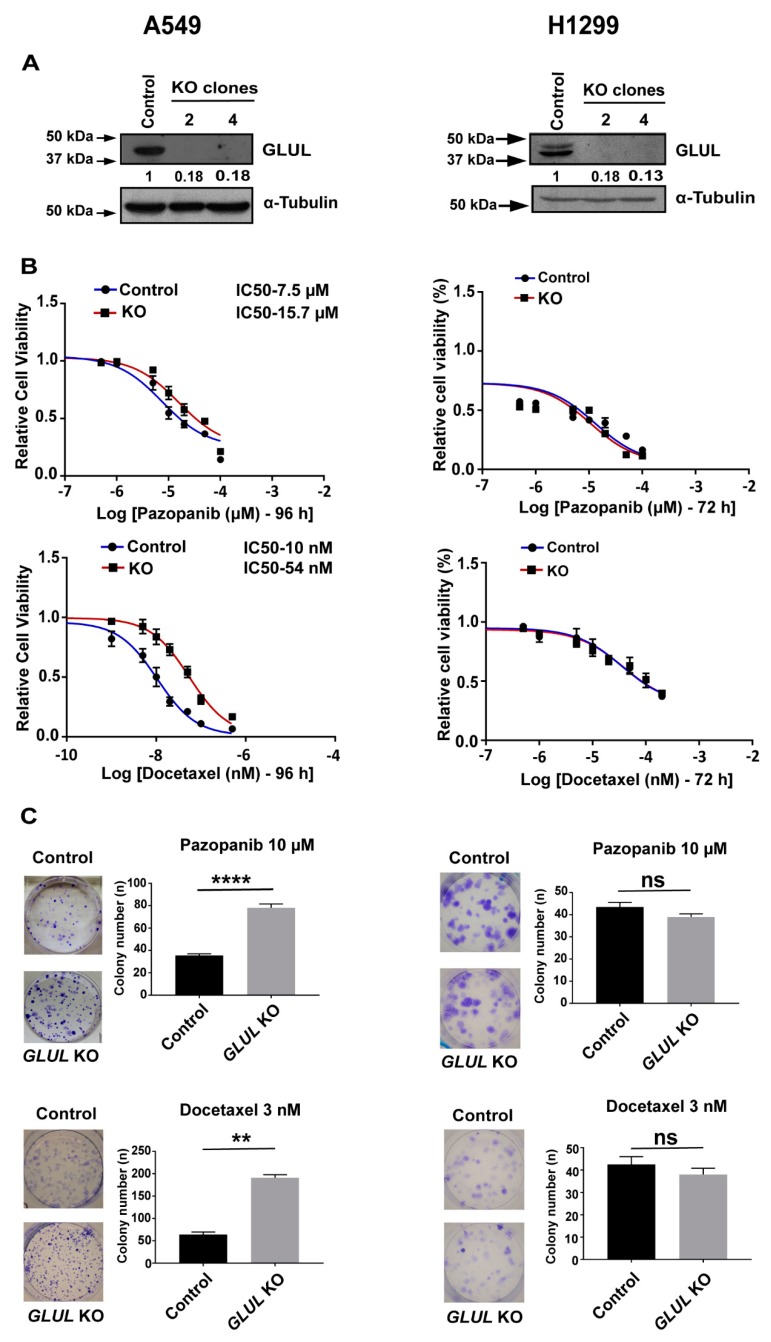
CRISPR/Cas9-mediated *GLUL* KO (knockout) conferred drug resistance in A549 but not in H1299 (**A**) A549 and H1299 cells were transfected with either negative controls (Lentiviral-empty vector) or Lentiviral CRISPR/Cas9-*GLUL* guide RNAs. Subsequently, cell lysates were analyzed by western blot (WB, see Methods) for GLUL expression, and individual clones devoid of GLUL expression were selected. The western blot membranes were subsequently probed with an anti-tubulin antibody to assess equal loading. The presence of GLUL and tubulin proteins are indicated on the right side of each blot. The signals for the GLUL and α-tubulin proteins were quantified by densitometry, and the numbers below GLUL blots indicate levels of GLUL protein in each lane following normalization of the signals with tubulin levels. For comparison, the signal intensity of GLUL in the control lane was assigned an arbitrary value of 1. The approximate location of various molecular weight markers is indicated on the left side of each blot. kDa, kilo Dalton. (**B**) A549 and H1299 control (DMSO-treated) or *GLUL* KO cells were treated with the indicated chemotherapeutic drugs, and the cell viability was analyzed by MTS assay after 72–96 h. IC_50_ for each drug was determined as indicated. (**C**) A549 and H1299 control and *GLUL* KO cells were treated with the indicated drugs and subjected to clonogenic assay. Histograms represent a total number of colonies counted, and the representative images of crystal violet stained cells are shown. The standard error (SE) bars in cell viability assay and clonogenic assay represent the means of three independent experiments. The data are shown as mean ± SEM; *p*-values were determined using a two-tailed unpaired t-test; **** *p* ≤ 0.0001; ** *p* ≤ 0.004, * *p* ≤ 0.01, ns—not significant.

**Figure 3 cancers-11-01945-f003:**
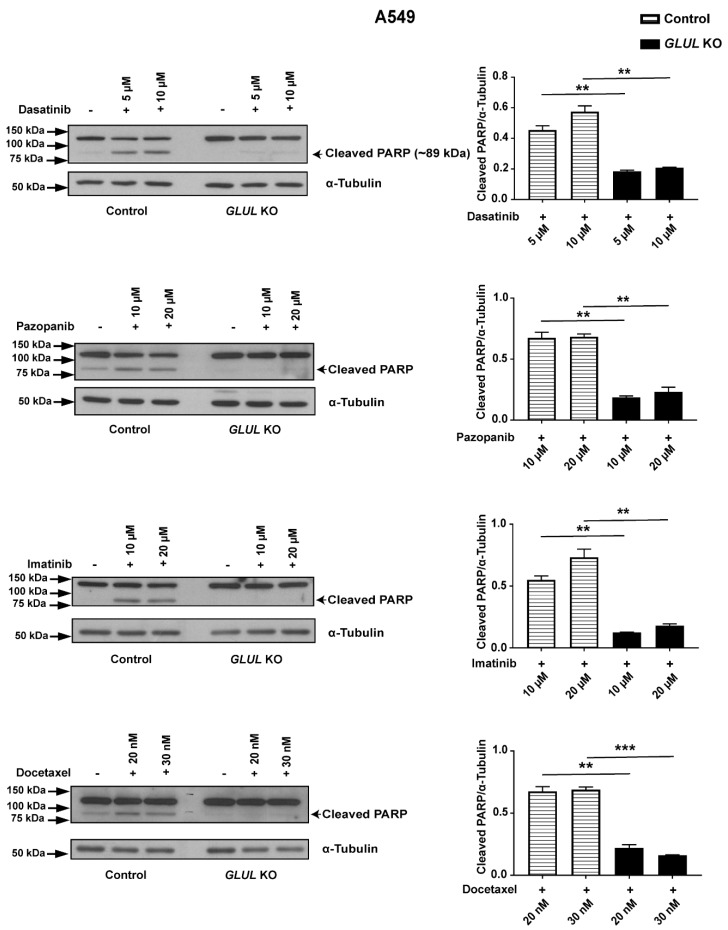
*GLUL* KO drug-resistant cells displayed reduced apoptosis. A549 control (DMSO-treated) or *GLUL* KO (cl3) cells were treated with various chemotherapeutic drugs for 12–18 h, as indicated. Subsequently, the expression of poly(ADP-ribose) polymerase (PARP)/cleaved PARP and α-tubulin were analyzed by western blot. Quantification of protein levels is presented on the ride side of each western blot, and the approximate location of various molecular weight markers is indicated on the left side of each blot. kDa, kilo Dalton. The data are shown as mean ± SEM; *p*-values were determined using a two-tailed unpaired t-test; ** *p* ≤ 0.001.

**Figure 4 cancers-11-01945-f004:**
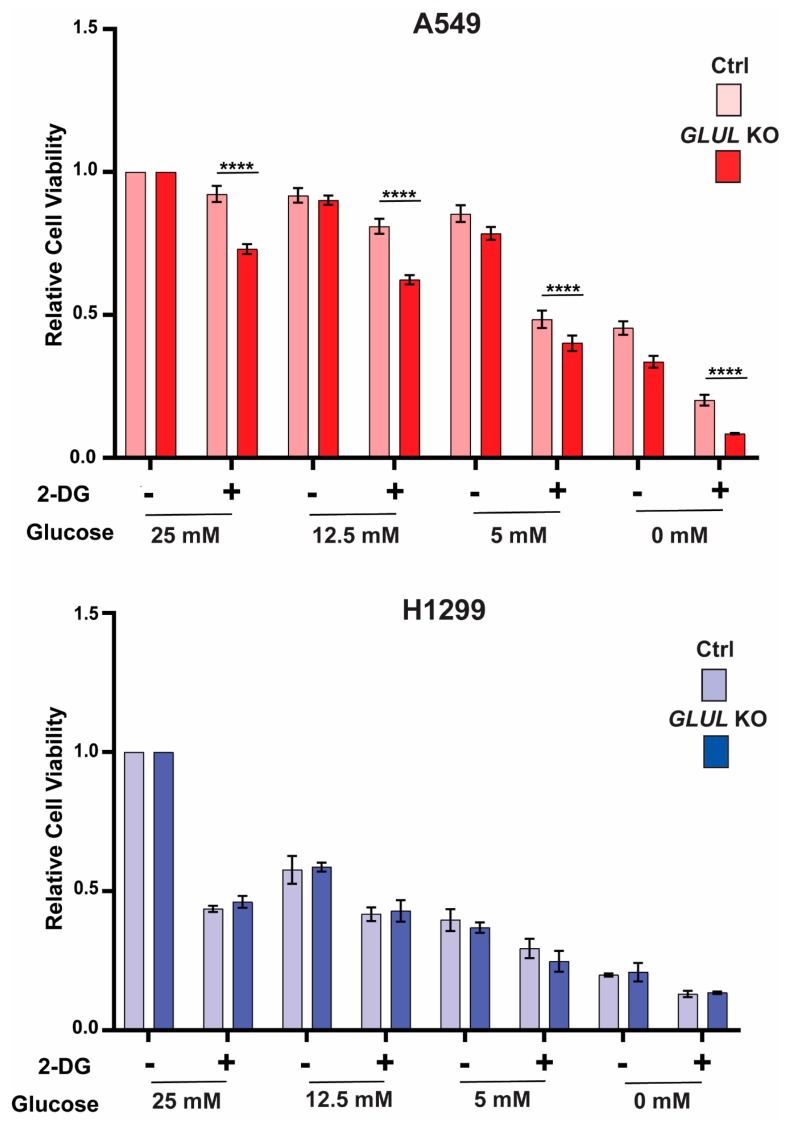
A549 *GLUL* KO cells displayed sensitivity to glucose deprivation. A549 and H1299 (control and *GLUL* KO) cells were cultured in various concentrations of glucose as indicated and were treated with 5 mM of 2-DG for 72 h, and the cell viability was analyzed by MTS assay. Data for each pair of cell lines (control/*GLUL* KO) was normalized to the data for the highest concentration of glucose (25 mM), as shown above. The data are shown as ± standard deviation. The standard deviation (SD) bars in the cell viability graph represent the means of three independent experiments. The data are shown as mean ± SD; *p*-values were determined using a two-way ANOVA; **** *p* ≤ 0.001.

**Figure 5 cancers-11-01945-f005:**
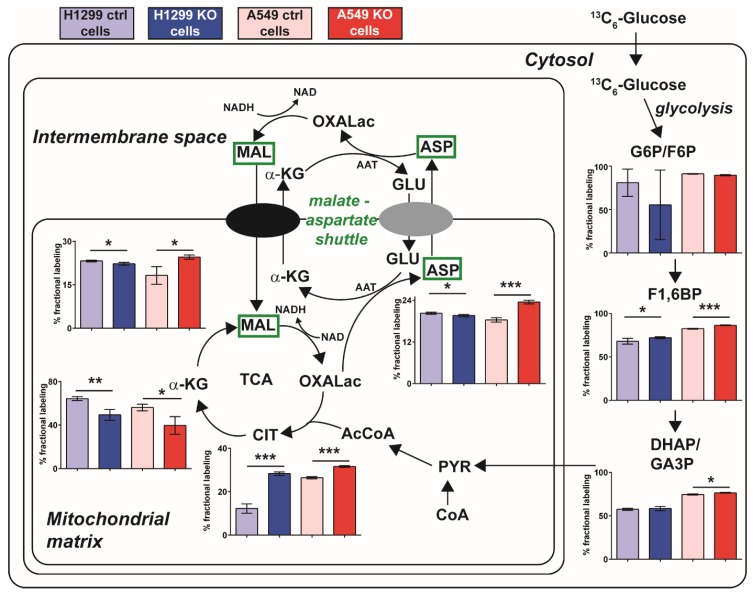
Increased flux through the malate-aspartate shuttle mediated drug resistance in A549 *GLUL* KO cells. Schematic illustration of a cell with mitochondria (intermembrane space + mitochondrial matrix). Superimposed are %-fractional labeling for various metabolic intermediates; glucose-6-phosphate/fructose-6-phosphate (G6P/F6P), fructose-1,6-bisphosphate (F1,6BP), dihydroxyacetone phosphate/glyceraldehyde-3-phosphate (DHAP/GA3P), citrate (CIT), α-ketoglutarate (α-KG), malate (MAL), aspartate (ASP), glutamate (GLU), oxaloacetate (OXALac), acetyl-coenzyme-A (AcCoA), pyruvate (PYR), coenzyme-A (CoA). Aspartate aminotransferase (AAT). Significance using unpaired students t-test * (<0.05), ** (<0.005), *** (<0.0005).

**Figure 6 cancers-11-01945-f006:**
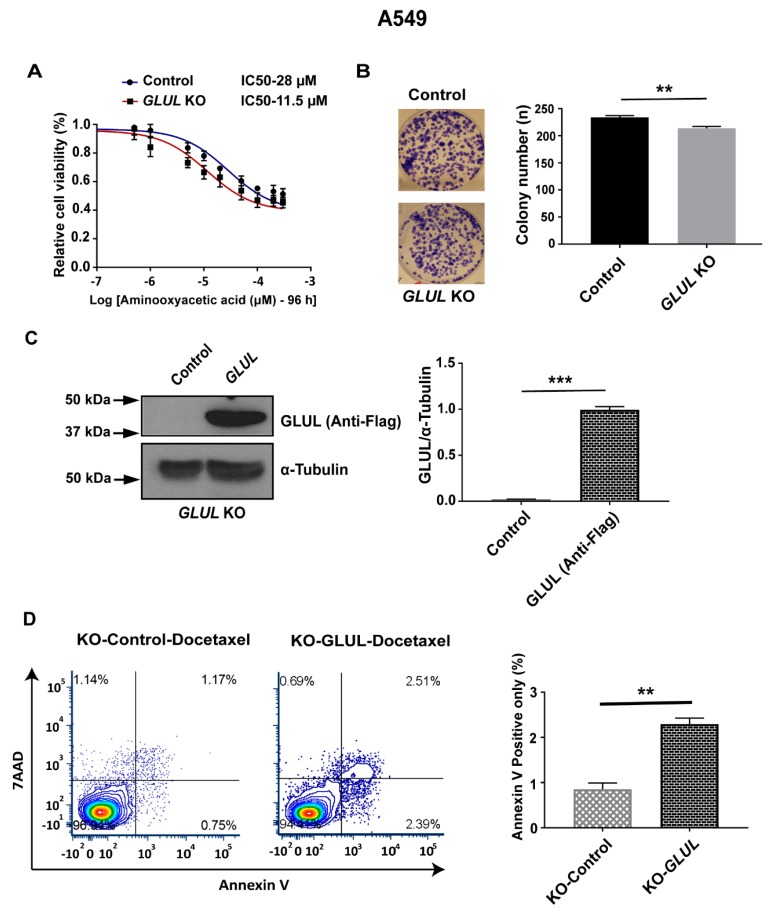
Targeting the malate-aspartate shuttle or *GLUL* expression re-sensitized drug-resistant A549 *GLUL* KO cells. (**A**) A549 *GLUL* KO/control cells (DMSO-treated) or treated with inhibitor aminooxyacetic acid at the concentrations were indicated. The cell viability was analyzed by MTS assay, and the IC_50_ values were determined as noted above (negative control (28 µM)—95% confidence interval (CI)-2.245e-005 to 3.591e-005 and *GLUL* KO (11.5 µM)—95% CI-8.416e-006 to 1.581e-005). The standard error (SE) bars in the cell viability curve represent means of three independent experiments. (**B**) Both control and *GLUL* KO cells were treated with the mentioned dose of aminooxyacetic acid for 12 days and subjected to colony formation assay. The column in the histogram represents the total number of colonies counted, and the colonies were photographed. (**C**) For *GLUL* overexpression, Flag-pcDNA3.1-*GLUL* cDNA clone and Flag-pcDNA3.1-empty vector were transfected in KO cells, and after 72 h, the cell lysates were analyzed by western blotting (WB) as in methods for GLUL expression levels, as shown above. The western blot membranes were subsequently probed with an anti-tubulin antibody to assess equal loading. The presence of Flag-tag-GLUL protein is indicated on the right side of the blot, and the quantification of protein levels are represented. The approximate location of various molecular weight is indicated on the left side of each blot. kDa, kilodalton. (**D**) Flow cytometric analysis of apoptosis assay; A549 *GLUL* KO cells expressing Flag-pcDNA3.1-empty vector were treated with DMSO (control), and Flag-pcDNA3.1-*GLUL* were treated with docetaxel (20 nM) treatment for 12 h, and the cells were stained with 7-Aminoactinomycin D (7-AAD) and annexin V. The % of annexin V positive cells were measured and represented. The standard error (SE) bars, in the apoptosis assay, represent the means of two independent experiments. The data are shown as mean ± SEM; *p*-values were determined using a two-tailed unpaired t-test; ** *p* ≤ 0.01, *** *p* ≤ 0.001.

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
