# Peer review of "GLUL Ablation Can Confer Drug Resistance to Cancer Cells via a Malate-Aspartate Shuttle-Mediated Mechanism"

_cancers, 2019, doi:10.3390/cancers11121945_

Round 1

Reviewer 1 Report

Authors decided not to show data on MDR1, PCNA, SLC1A5 (previously supplementary fig. S5) and not to adjust the conclusions in the same time. That further undermines the credibility of the discussion of this part of the research. I recommend leaving the relevant figure in supplement but adjust the text appropriately

Author Response

We have brought back the suplementary figure showing the proteins reviewer is mentioning. It is now Figure S7. The text have been updated accordingly. In addition, as the reviewer asked for minor language editing. One of the co-authors who is native speaker of English has made several language editions.

Reviewer 2 Report

Dear Editor,

I do not have any further suggestions. All my questions were answered. 

Kind regards,

Author Response

Ok, thank you

This manuscript is a resubmission of an earlier submission. The following is a list of the peer review reports and author responses from that submission.

Round 1

Reviewer 1 Report

In the submitted manuscript, Muthu et al., describe that ablation (siRNA, shRNA, CRISPR) of glutamine synthetase (glutamate-ammonia ligase, GLUL) may be a mechanism leading to drug resistance in some neoplastic cells. The GLUL downregulation facilitates the malate-aspartate shuttle making the cancer cells more viable.

The 6 different cell lines were initially chosen for checking the GLUL knock-down on their viability and one (ALL) was previously attributed with GLUL downregulation. Is there any genetic (mutation?) or metabolic common background that might have been suspected/deducted in advance to be involved in GLUL-mediated effects? In other words: were the cell lines of different origin chosen randomly? Can there be any conclusion drawn as to the common link making the particular neoplasms susceptible to GLUL ablation?

2. For the reader convenience authors might include information that A549 and H1299 are of the same origin (NSCLC), what makes the latter a proper control.
I strongly recommend to move H1299 experiments from the supplement (Fig.S1) to the manuscript with additional experiment (clonogenic asssay) to make the results parallel to A549.

The shRNA results doubling the CRISPR method are less important here and may be moved down in description.

(The same issue with H1299 and PARP cleavage – additional experiment would be nice but not mandatory).

The mechanism of action of each drug and the reasoning for its use in a particular experiments (e.g. 5 in viability test and 3 out of 5 in colony test) should be described in a separate paragraph.

Fig.1 B: The control level of GLUL oscillates between 0.5 – 1.5 with relatively small SEM. The all control levels are pretty far from one. How was the data calculated?

‘An increased flux in the malate aspartate shuttle could work together with increased glucose utilization towards increasing the metabolic fitness through increased capacity of NADH production’

‘These findings support a mechanistic model where GLUL ablation increases both glycolytic flux and the malate-aspartate shuttle concurrently. This increases the metabolic fitness of cells, which aides cell survival under drug pressure.’ On the other hand, ‘Labelling of the glycolytic intermediates showed similar trending patterns for both cell lines as well as the TCA intermediates α-ketoglutarate and citrate (Figure 5).’ In the same time H1299 (Fig.4) seem to have relatively higher than A549 glucose and glutamine dependence. This baseline difference in glucose/glutamine dependence correlates with no effect of GLUL depletion. The differences shown in Fig.5 for glycolysis are very small. Is there a possibility that malate/aspartate operates differently in GLUL KOs, and therefore (and not additionally) more glucose is needed?

I recommend to provide the results of AAT inhibition in H1299 cells (Fig.6) – it would be neat to see no effect on their viability. Does AAT decrease basal viability in control cells?

The results from 2.7 paragraph should be presented as a separate figure.

FIG S5 – conclusions in Discussion are based on a single experiment? If there are more repeats, it could be moved to the manuscript. Otherwise, the statement should be mitigated

The siRNA procedure should be described in more detail in M&M section.

Line 63: delete ‘recently’

Line 217 delete ‘were found’

Author Response

Please see the attchement

Reviewer 2 Report

Dear Editor,

Pleased find my review of the manuscript entitled: “GLUL ablation can confer drug resistance to cancer cells via a malate-aspartate shuttle-mediated mechanism" by Magesh Muthu et al.

The paper is clearly written and the figures are clear. The paper per se is very engaging but my enthusiasm for this study was highly decreased by the use of one cell resistant and one sensible in the majority of the experiments. Therefore, additional experiments and extra information are required in order for it to be appropriate for publication in Cancer journal.

Mayor issues:

All the western blot must have the molecular weight. Figure 1B and figure 3. The authors said: “Interestingly, knocking down GLUL promoted drug resistance in two of the cell lines (A549 and HCC1954; Figure 1C). As GLUL KD induced the highest level of drug resistance in A549 cells but had no apparent effect in the NSCLC H1299 cells, we choose to compare these two cell lines further to identify potential mechanisms of the resistance.” This statement must be turn it down. Because unfortunately, Docetaxel resistance effect in A549 and HCC1954 is so small. I wonder, if the authors has a better and strong effect in cell proliferation? Unfortunately, my enthusiasm for this study was highly decreased by the use of one cell resistant and one sensible in the majority of the experiments. Therefore, some experiment must be validated in other cells. Other big issue for me is how the authors dissolved the chemotherapeutic drugs. The authors dissolved the drugs in DMSO. For example, in the clinic, cisplatin is typically provided as a lyophilized powder in a vial containing 50 mg cisplatin, 450 mg NaCl, and 500 mg mannitol. It has been demonstrated that DMSO could affect the effect of several drugs. Cancer Res. 2014;74(14):3913-22. Say no to DMSO: dimethylsulfoxide inactivates cisplatin, carboplatin, and other platinum complexes. Hall MD, Telma KA, Chang KE, Lee TD, Madigan JP, Lloyd JR, Goldlust IS, Hoeschele JD, Gottesman MM. It would be nice if the authors validate the effect of glucose and glutamine withdrawal in GLUL WT and KO A549 and H1299 cells with the treatment with 2-deoxy glucose (2-DG) and 6-Diazo-5-oxo-L-norleucine (DON). The authors said: “Therefore, we hypothesized that GLUL ablation could mediate an increased capacity of the malate-aspartate shuttle. Thus, even in the absence of chemotherapeutic drugs, GLUL KO cancer cells should be more dependent on support of this shuttle for mitochondrial NADH production and ultimately respiration and therefore more sensitive to inhibition of the shuttle system.” But the authors did not measured mitochondria oxygen consumption, the authors also need to treat the cells with rotenone (a complex I inhibitor) and show that the GLUL KO cancer cells are more sensible.
